# Ontology-Driven Semantic Analysis of Tabular Data: An Iterative Approach with Advanced Entity Recognition

**Madina Mansurova, Vladimir Barakhnin, Assel Ospan \***  **and Roman Titkov**

Faculty of Information Technology, Department of Artificial Intelligence and Big Data, Al-Farabi Kazakh National University, Almaty 050040, Kazakhstan; madina.mansurova@kaznu.edu.kz (M.M.)
* Correspondence: asselyaospan@gmail.com; Tel.: +7-708-151-22-39

**Abstract:** This study focuses on the extraction and semantic analysis of data from tables, emphasizing the importance of understanding the semantics of tables to obtain useful information. The main goal was to develop a technology using the ontology for the semantic analysis of tables. An iterative algorithm has been proposed that can parse the contents of a table and determine cell types based on the ontology. The study presents an automated method for extracting data in various languages in various fields, subject to the availability of an appropriate ontology. Advanced techniques such as cosine distance search and table subject classification based on a neural network have been integrated to increase efficiency. The result is a software application capable of semantically classifying tabular data, facilitating the rapid transition of information from tables to ontologies. Rigorous testing, including 30 tables in the field of water resources and socio-economic indicators of Kazakhstan, confirmed the reliability of the algorithm. The results demonstrate high accuracy with a notable triple extraction recall of 99.4%. The use of Levenshtein distance for matching entities and ontology as a source of information was key to achieving these metrics. The study offers a promising tool for efficiently extracting data from tables.

**Keywords:** semantic analysis; OWL ontology; table interpretation; knowledge triplets; entity classification; Levenshtein distance



## 1. Introduction

Currently, data collection and analysis are an integral part of the vast majority of fields of activity. Information can be presented in various formats, such as text, tables, and images. At the same time, one of the critical tasks is the analysis of tables that contain large amounts of information.

To extract the information needed from a table, one needs to analyze it and understand the semantics, including the content type of each cell and the relationships between them. It is important to consider that the structures of tables can differ significantly from each other.

Understanding the semantics of a table makes it possible to find relationships between various data, form a general picture of the contents of the table, and extract information in a convenient form for further processing.

The aim of the work is to develop and implement a technology for the semantic analysis of tables using OWL ontology [1]. OWL is a formal knowledge description language used in semantic web applications. It is used to create ontologies that describe relationships between entities and concepts in a particular field of knowledge.

To further elucidate and achieve this overarching goal, several pivotal research questions have been formulated:

1. How can the semantics of tables be effectively understood to facilitate accurate data extraction?
2. How can ontology, particularly OWL ontology, be leveraged to enhance the semantic analysis of tables?

3. What is the efficacy of the proposed iterative algorithm in parsing table contents and determining cell types based on ontology?
4. How do advanced techniques like cosine distance search and neural network-based table subject classification augment the efficiency of our method?
5. How does our method compare in terms of versatility and adaptability to existing table extraction methods, especially in terms of language and field specificity?

In the proposed research, we will try to answer these questions and provide evidence.

To solve the problem of semantic analysis of tables and extract information, an iterative algorithm is proposed that performs a primary analysis of the table and determines the content type of each cell: a number or a string; for each cell containing a string, the algorithm analyzes the value of this cell and specifies its type: the cell can be an object from the ontology, a property or relationship between objects, or a third-party reference. If the cell value is an object, then the row containing that cell is marked as the row containing the found object; if the cell value is a property or relation, then the column containing that cell is marked as the column containing that property or relation; for each cell containing a number, its row is checked for an ascending sequence, which is a sequence of years. If such a sequence is found, the algorithm considers that the rows of the table should contain objects, the columns should contain years (or other consecutive time intervals), and each cell is the value of a property or object, the type of which is determined using the table subject classifier; then for each cell whose row contains an object and whose column is a property or relation, a new triplet is formed, consisting of the object, property or relation, and the value of the cell; each formed triplet is added to the ontology, which allows for a more accurate analysis of subsequent tables.

The scientific novelty of this work lies in the creation of a new (automated) method for extracting data from tables in any language in an arbitrary field of knowledge—subject to the availability of an ontology that describes the data structure. Approaches are also used to classify data more accurately (such as search by cosine distance and a neural network to classify the subject of a table) in comparison with similar algorithms.

While there have been various methods developed for data extraction from tables, most are either language-specific or field-specific. Our method's versatility in handling tables in any language and from any field of knowledge is a distinguishing feature. For instance, while the work by [2] provides techniques for data extraction in machine learning papers, chemistry tables, and webpages, it does not generalize to other contexts as our method does.

The reliance on ontologies to describe the data structure is another unique aspect of our approach. Prior works, such as those by [3], have employed rule-based or heuristic methods for data extraction without the need for an external ontology. Our method's ability to adapt to different data structures by simply swapping the ontology adds a layer of flexibility absent in many existing algorithms.

Advanced Data Classification Techniques: The combination of cosine distance search and neural network-based classification for determining the subject of a table is a significant advancement. While individual components like cosine distance search have been employed in prior works (e.g., [4]), the synergistic combination of these techniques in our approach has shown better performance in our tests.

As a result of the work, a software application was created for the semantic classification of tabular data, which takes a table and an ontology as input and as output data produces the objects found in the table and their attributes based on information about the structure of the subject area in the ontology.

The result obtained makes it possible to significantly speed up the transfer of information from tables to ontology in order to solve the problems of summarizing statistics and finding patterns between data.

To achieve the goals set, the following tasks were defined:

1. Designing the architecture of the module and its components.
2. Development of an OWL ontology.

3. Development of a tabular data processing module.
4. Implementation of the tabular data classification module.
5. Implementation of data export in the form of triplets.

Let us describe the structure of the article. First, an analysis of existing approaches for the English language is carried out, and requirements for the program are drawn up. Next, the key approaches used to implement the algorithm are described, as well as the technologies used are described and justified. Then, the implementation of the algorithm is described in detail, what difficulties were encountered and how they were solved. Finally, the methods for testing the program and the test results are described, and the accuracy of the developed algorithm is also evaluated.

## 2. Related Works

Nowadays, there are several solutions that allow for the semantic analysis of tables.

TableMiner+ [5] is a tool designed to automatically extract table structures and content from HTML documents. It detects tables, extracts headers, cells, columns, and rows, and determines the data types in tables. The tool uses an iterative approach to refine the quality of data extraction. However, it may run into problems when working with complex tables, such as tables with graphics or formulas. TableMiner+ works based on certain table formatting rules and requires adjustments for tables that do not follow those rules. It only handles PDF and HTML formats. Despite its limitations, it is effective for semantic data extraction from tables and has applications in areas such as medical statistics and data science.

QTLTableMiner++ [6] extracts semantic information from tables in articles related to genetics. It uses natural language processing and machine learning to extract information about genetic variants from tables. The tool defines tables of genetic variant data and links them to the corresponding genetic loci. It is based on the Apache Solr search engine and can index and search text data in multiple languages. However, its use is limited to certain table formats from a particular resource. Overall, it is a powerful tool for extracting genetic data, but it has limitations, such as a limited table format and a domain-specific codebase.

Qurma is a pipeline for extracting tables, interpreting and enriching knowledge graphs from various sources, based on the Camelot library [7]. Qurma efficiently processes HTML pages, PDFs, and images initiated by a user-provided document URL to retrieve a table. The result is an optimized dataset that can be exported as CSV, JSON, or attribute values.

Qurma's conceptual architecture is based on the Clean Architecture paradigm, with a focus on controlled interfaces and a layered structure.

MantisTable is a public tool for semantic annotation of tables from scientific publications [8]. It identifies objects in table cells and links them to knowledge bases like DBpedia using string matching, word embedding, and machine learning. The tool was evaluated on 100 tables from scientific articles, comparing its abstracts with those of experts. It demonstrated high F1 values for object recognition and linking. Compared to other tools, MantisTable showed the best F1 values for both tasks. Its intuitive interface is highly acclaimed by users. Semantic annotations from MantisTable can improve the search and retrieval of scientific content.

Berners-Lee et al. introduced the concept of the Semantic Web, where data is given a well-defined meaning that allows computers and people to work together [9]. OWL is a key component of this vision, providing a rich language for defining structured web ontologies. The disadvantages of this method are (1) difficult to implement, as the vision of the Semantic Web requires significant changes to the existing web infrastructure, and (2) dependence on widespread adoption; for the Semantic Web to be effective, most web content creators must adopt standards, (3) the possibility semantic inconsistencies; different creators may interpret values differently, resulting in semantic inconsistencies.

Hurst, M. discussed the problems and techniques involved in interpreting tables in documents, emphasizing the need to understand both structure and semantics [10]. It has

the following disadvantages: (1) limited to document-based tables—may not be efficient for web tables or dynamic tables; (2) problems handling different table structures and formats.

In [11], the authors proposed an approach to interpreting tables using linked data and ontologies. Their method focuses on creating RDF triplets from tables. However, this method has a dependency on the quality and completeness of ontologies and potential problems when matching tables with existing ontologies, especially if they are not aligned.

The authors of [12] developed integrated machine learning with semantic technologies to interpret and annotate tables where there are difficulties with dependence on training data since the quality and quantity of training data can affect performance.

A tool called Table2OWL has been developed to transform tables into OWL ontologies. It provides a semi-automatic way to convert tabular data to semantic web formats [13], where manual intervention is required, which can be time-consuming, and errors are possible when converting tables in OWL ontologies. Elaborating further, the complexity arises due to diverse data structures in tables, requiring precise class and property decisions in OWL. Often, tables possess inconsistencies or ambiguous data, which, when mapped to ontologies, can lead to representation errors. Moreover, preserving the semantic integrity of information during conversion demands expert judgment, especially when interpreting relational nuances from tables. Post-conversion, thorough validation against the source table is indispensable. Despite Table2OWL's assistance, manual oversight, driven by domain expertise, remains crucial to ensure accuracy.

Limaye et al. introduced a system of semantic table annotation, focusing on linking table cells to DBpedia objects [14]. The disadvantage of this method is the limitation of binding to DBpedia objects: it may not cover all possible semantic annotation, and problems with handling ambiguous or obscure table cell contents.

Poggi et al. discussed data access based on OBDA ontologies, where relational data sources are practically integrated with the ontology, providing a semantic representation of data [15]. Here, the authors face the difficulty of integrating relational data sources with ontologies.

Another study [16] presented WebTables, a system that retrieves and indexes tables from the Internet, providing a semantic search of a huge amount of tabular data. However, there are also problems with retrieving and indexing various web tables, and there may be semantic inconsistencies during searches.

Bhagavatula et al. proposed methods for linking tables to the correct data models, which allows better understanding and querying of tabular data [17]. The work has been successfully applied to tables but faces the following difficulties: (1) dependency on existing data models; if the models are not exhaustive, interpretation can be limited; (2) problems in associating different tables with the correct models.

To address the problems of interpreting tables, Venetis et al. discussed the problems of interpreting tables, especially when working with web tables, which can be noisy and heterogeneous [18]. Although this work highlights the problems, it does not offer comprehensive solutions.

*System Analysis of Presented Solutions*

Thus, the developed algorithm should combine the advantages of existing solutions but not have their disadvantages. The main feature of the solution should be the versatility of its application. To do this, a modular approach is proposed—for each file type (XLSX, CSV, PDF), an appropriate processing module is created that will extract the table from the file and pass it on for analysis. This approach makes it easy to add support for new file types.

An important requirement resulting from universality is the ability to process tables in an arbitrary area of knowledge. To this end, the module must use an OWL ontology that describes the structure of the knowledge area. It lists the main types of objects, their properties and relationships between them. This approach will allow processing tables in any area of knowledge, provided that the created OWL ontology is available.

When using ontologies in table analysis, tables can be mapped to specific classes and properties described in the ontology. For example, if a table contains information about different types of animals, then this table can be associated with the Animals' ontology class, and the columns of the table can be associated with the corresponding properties, such as "species" or "habitat".

Approaches using ontologies are an effective solution for data classification since these approaches provide additional information about the structure of a knowledge area. One example of the use of ontologies is the algorithm described in [19], which uses an ontology to train a neural network in order to perform a semantic analysis of tables and create knowledge graphs based on data from a table. This solution has a high classification accuracy but requires retraining of the algorithm when working in a new area of knowledge.

Ontological approaches to the semantic analysis of tables have several advantages. They provide the opportunity to obtain a higher accuracy of analysis since ontologies provide a formal way to define the conceptual relationships between objects and concepts. In addition, ontologies can be used to automatically create rules for parsing tables, which reduces the need for manual rule creation.

However, ontology-based approaches also have disadvantages. For example, ontologies can be complex and require significant time and effort to create and maintain. In addition, the use of ontologies can be limited to the subject area for which the ontology was created, which makes the approach based on this ontology less universal.

In general, ontology-based approaches can be useful in cases where high accuracy of analysis is required and when an ontology already exists or can be created for the subject area in which the program for semantic analysis of tables works. For example, there is a solution described in the article [20] which allows the creation of an ontology based on tabular data. After running the algorithm, this ontology will be automatically expanded, then properties, relations between objects, and additional classes of objects will be added.

Very often, even when using ontologies, one cannot avoid heuristic ideas about the structure and semantics of tables. If the solution uses such heuristic representations, then it relies on predefined rules to extract information from the tables. These rules can be manually created by subject matter experts or can be automatically extracted from large datasets using machine learning techniques.

Rules can describe various aspects of tables, such as structure, formatting, and cell content. For example, a rule might say, "if the first column of a table contains dates, then the second column must contain the corresponding values." non-standard tables or changes in the structure and formatting of tables.

Hybrid approaches to the semantic analysis of tables combine several methods and technologies to achieve more accurate and complete data extraction from tables. Hybrid methods inherently exhibit superior adaptability, accommodating varied document types and formats. While rule-based techniques might falter with unconventional table patterns, leading to potential misses or false positives, ontology-based approaches offer semantic depth, enabling accurate extraction even from less structured tables. Moreover, by cross-referencing with domain-specific knowledge, ontology-driven strategies bolster accuracy, minimizing errors typically seen with rule-based extraction. Consequently, the combined strengths of these methods in a hybrid approach enhance both adaptability and precision in data extraction.

In this paper, to conduct semantic analysis in the developed algorithm, it was decided to use a hybrid approach based on a combination of rules and ontologies. When using this approach, rules are first defined for extracting the structure of the table and data from it; then these data are associated with the corresponding concepts in the ontology, which determine their semantics. For example, if a table contains weather data for different cities, data extraction rules can be defined based on the columns "city", "temperature", "pressure", etc., and then these data are associated with the appropriate classes in the ontology, such as like "City", "Temperature", "Pressure", etc.

Hybrid approaches can be more effective than separate methods because they combine the advantages of each method and achieve more accurate and complete data extraction from tables.

## 3. Materials and Methods

### 3.1. Requirements for the Semantic Analysis of Tables

After analyzing the existing solutions and formulating the distinctive features of the program, we can single out a list of the main requirements for the program for the semantic analysis of tables:

1.  Support CSV, XLSX and PDF file formats for processing. The program must be able to process tables that are presented in various file formats: CSV, XLSX and PDF (see Table 1). These formats are the most common in use for storing and exchanging data in tables.
2.  Ability to easily add the ability to handle new file formats. The requirement suggests that the program must be able to extend and support the processing of new file formats. This means developers can easily add support for new file formats if needed.
3.  Automatic removal of empty rows and columns of the table. The program should automatically remove empty rows and columns from the table. This will simplify the processing process and reduce the amount of data that does not carry useful information.
4.  Ability to process tables in an arbitrary area of knowledge. This requirement means that the program must be able to process data-themed tables. This is important because the ability to process tables in an arbitrary knowledge area is a key condition for product scalability. In addition, this requirement will allow for the creation of a module that is significantly superior to analogues, since other solutions are usually designed to work in one highly specialized area of knowledge.
5.  Using an OWL ontology with a description of the data structures to be extracted. The requirement suggests that the program must use the OWL ontology to determine the types of data to be retrieved from the table. Modules also receive information about existing objects, properties and relationships between them from the ontology. This will allow the program to correctly identify and extract the required data.
6.  The possibility of exporting results to ontology in OWL format. After the analysis of the table is completed, the program should be able to export the results of processing the table into an ontology in the OWL format. This will make it easy to analyze and use the data obtained. At the same time, it should be possible to supplement the original ontology so that when processing subsequent tables, the module can use data from previous tables.
7.  Processing large amounts of data. The program must be able to process large amounts of data. This may be required if many tables or tables with many rows and columns need to be processed.

**Table 1.** Example of a table that fits rule 2.

| Population_KZ | | | | | | |
|---|---|---|---|---|---|---|
| **Regions_KZ** | **2016** | **2017** | **2018** | **2019** | **2020** | **2021** |
| Kazakhstan | 17,669,896 | 17,918,214 | 18,157,337 | 18,395,567 | 18,631,779 | 18,879,552 |
| Akmola_region | 744,420 | 734,369 | 738,942 | 738,587 | 736,735 | 735,566 |
| Aktobe_region | 834,808 | 845,679 | 857,711 | 869,637 | 881,651 | 894,333 |
| Almaty_region | 1,947,552 | 1,983,465 | 2,017,278 | 2,038,935 | 2,055,724 | 2,077,967 |
| Atyrau_region | 594,511 | 607,528 | 620,684 | 633,791 | 645,280 | 657,110 |
| West_Kazakhstan_region | 636,980 | 641,513 | 646,927 | 652,325 | 656,844 | 661,316 |
| Zhambyl_region | 1,110,749 | 1,115,307 | 1,117,218 | 1,125,440 | 1,130,099 | 1,139,192 |
| Karaganda_region | 1,384,810 | 1,382,734 | 1,380,537 | 1,378,532 | 1,376,882 | 1,375,938 |
| Kostanay_region | 883,806 | 879,134 | 875,616 | 872,795 | 868,549 | 864,550 |

**Table 1.** *Cont.*

| | Population_KZ | | | | | |
|---|---|---|---|---|---|---|
| **Regions_KZ** | **2016** | **2017** | **2018** | **2019** | **2020** | **2021** |
| Kyzylorda_region | 765,058 | 773,143 | 783,157 | 794,335 | 803,531 | 814,588 |
| Mangistau_region | 626,774 | 642,824 | 660,317 | 678,199 | 698,796 | 719,571 |
| Pavlodar_region | 758,594 | 757,014 | 754,854 | 753,853 | 752,169 | 751,012 |
| North_Kazakhstan_region | 569,594 | 563,300 | 558,584 | 554,517 | 548,755 | 543,735 |
| Turkestan_region | 2,840,871 | 1,966,336 | 1,977,028 | 1,983,969 | 2,016,037 | 2,044,742 |
| East_Kazakhstan_region | 1,396,019 | 1,389,568 | 1,383,745 | 1,378,527 | 1,369,597 | 1,363,797 |
| Astana | 872,584 | 972,692 | 1,030,577 | 1,078,384 | 1,136,156 | 1,184,411 |
| Almaty | 1,702,766 | 1,751,308 | 1,801,993 | 1,854,656 | 1,916,822 | 1,977,258 |
| Shymkent | - | 912,300 | 952,169 | 1,009,085 | 1,038,152 | 1,074,466 |

### 3.2. Algorithm

The sequence of the algorithm is as follows:

1. Read the table from a file and process it with a module corresponding to the file type (CSV, XLSX, PDF).
2. Conduct a primary analysis of the table and determine the content type of each cell ("Number" or "Text", "Date").
3. For each line, check for an increasing sequence of dates.
4. For each cell, the content of which is "Text", analyze the value and specify its type (object, property or relation).
5. If the cell value is an object, mark the line containing this cell as the line containing the found object.
6. If the cell value is a property or relationship, mark the column containing that cell as the column containing that property or relationship.
7. If a sequence of years or dates was found in the table, then it is considered that the rows of the table should contain objects, the columns—years or dates, and each cell is the value of a property or object, the type of which is determined using the table subject classifier.
8. If a sequence of years or dates is not found in the table, then for each cell whose row contains an object and whose column is a property or relation, form a new triplet consisting of the object, property (or relation) and cell value.
9. Add the generated triplet to the ontology.
10. Repeat the algorithm from step 7 for each table cell.

The sequence diagram of this algorithm is shown in Figure 1.

Next, we describe the approaches used to develop and implement the above algorithm.

### 3.3. Rules for Defining Table Structure

For the developed algorithm, several rules were identified that help determine the structure of the table:

1. "If the first row of the table contains dates, then the rows should contain objects. Such a table reflects the statistics of one property of an object for various timestamps.
2. There is also a special case of rule 1, which must be handled separately: "If the first row of the table contains a sequence of integers that are a sequence of years, then the rows must contain objects. Such a table reflects the statistics of one property of an object for various timestamps. For example, a table describing the statistics of the population of regions by years has a similar structure (see, for example, Table 1).

In general, rule-based approaches are useful in cases where the structure and formatting of tables are known in advance and where high precision in extracting information is required. They can also be effective in combination with other methods, such as machine learning methods or pattern-based approaches.

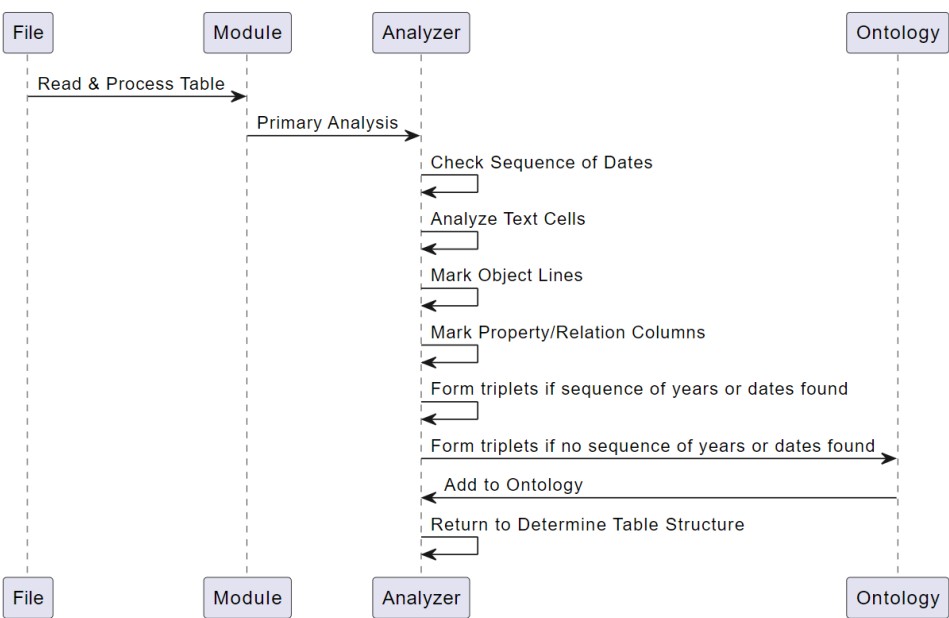

**Figure 1.** Sequence diagram for semantic analysis of tables and obtaining triplets for ontology.

*3.4. Codebase Independence from Knowledge Area*

The main difference between the developed algorithm and the analogs discussed above is that the algorithm has the ability to work in an arbitrary area of knowledge, while similar solutions can only work in predetermined areas of knowledge on which they were trained or which were incorporated into the logic of the algorithm. So, for example, QTLTableMiner++ can process tables in the field of genetics, and at the same time, information about the structure of the knowledge area is hard-coded into the code. This does not allow the application of this algorithm in other areas of knowledge. Therefore, an approach was proposed, the code base of which does not contain any information about the structure of the knowledge area.

An OWL ontology is used to describe the knowledge area—it contains information about the hierarchy of objects, their properties and relationships. OWL ontology is great for describing the semantic relationships between objects, allowing for objects and concepts to be semantically linked to each other. Thus, in order for the solution to be developed to be able to process tables in an arbitrary knowledge area, it is sufficient to develop an ontology for this knowledge area and describe the basic concepts and objects that need to be extracted.

So, for example, to analyze the dependence of the economic state of the population on the availability of water resources, it is necessary to develop an ontology that will include the following types of objects: City, Region, River, Lake, and Sea.

For cities and regions, it is necessary to set the properties of interest to us, for example, the income of the population, population, and so on. For rivers/lakes/seas, the appropriate characteristics must be added, for example, depth and volume of water.

The final stage will be the creation of relationships between objects, which makes it possible to semantically relate objects to each other. For example, the relation "Located at" and its inverse relation "Contains" must be entered. Using these relationships, for example, cities, regions and water bodies can be linked. The resulting OWL ontology will reflect the structure of the knowledge area.

The resulting ontology will be used during the analysis of tables and filled with the found data.

**4. Software Implementation**

To implement this algorithm, the Python programming language, the Owlready2 library, and the Protégé program were chosen.

The Owlready2 library [21] is used to work with OWL ontologies in the Python programming language. It provides the ability to create, modify, load and save ontologies, as well as perform various operations with them, such as searching for class instances, checking instance properties, etc.

Protégé [22] is a free program for creating and managing ontologies that allow the structure of knowledge in a certain area to be modeled and described.

### 4.1. Development of OWL-Ontology in the Field of Monitoring Water Resources of Kazakhstan

The subject area of the ontology is "Water resources of the Republic of Kazakhstan". This data includes information about water resources: rivers, lakes, reservoirs, their main indicators: length/area, water level, water class, etc. The ontology also contains data on regions—the number of people in cities, regions, birth, and death rates. Figure 2 shows the schema of the ontology.

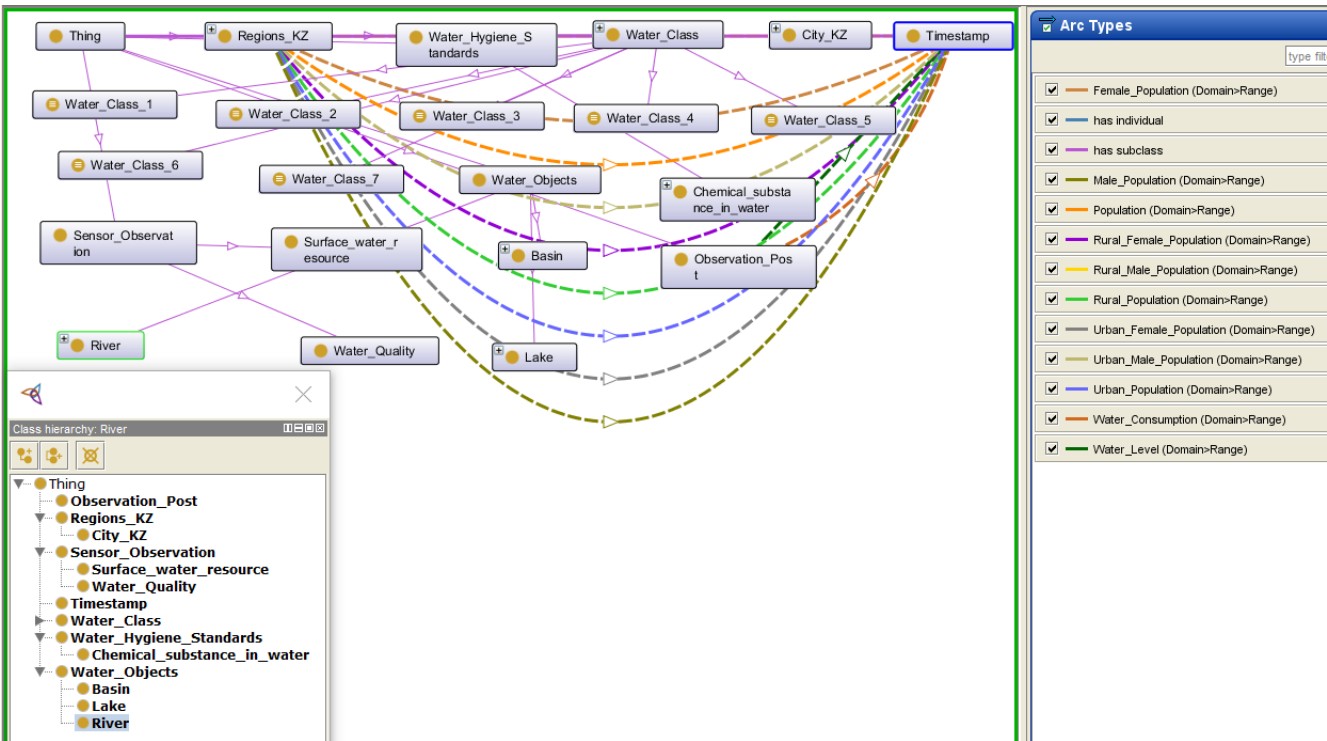

**Figure 2.** Scheme of the developed ontology on Protege.

The purpose of creating an ontology is to find links between the economic indicators of regions and water bodies. The ontology should help answer the question of how water resources affect the standard of living of the population.

The ontology contains two main classes of objects—Region and Water_Object. The regions are the regions of the Republic of Kazakhstan. In addition, cities can be located in regions—for this, a separate City class has been made, which is the successor of the Region class.

The Region and City classes have the following property structure:

1. Population
2. Male population
3. Female population
4. Urban population
5. Male urban population
6. Female urban population
7. Rural population
8. Male rural population

9.  Female rural population

Each property is timestamped with a Timestamp, allowing for the data by year, month, or even day to be entered. This allows for tracking the trends of a particular property of an object over time.

There is also a structure of relations between objects:

1.  Located_In—located in (reverse to Contains)
2.  Contains—contains (reverse to Located_In)

The Located_In and Contains properties are reciprocal—if a region contains a lake, then it must be true that the lake is located (Located_In) in the region.

The Water_Objects class includes 3 types of objects—Basin (pool), Lake (lake), and River (river). These water bodies are connected both with each other (for example, a river is in a basin) and with regions.

For water bodies, there is this property structure:

1.  Area (lake, basin)/Length (river).
2.  Water level.
3.  Class of water (according to the degree of pollution).
4.  Elevation difference (river).
5.  Water temperature.
6.  Location (latitude, longitude).
7.  Water resources.

The remaining object classes (other than Region and Water_Object) describe additional information needed in the analysis. All data are interconnected using a timestamp or coordinates, which allows for analysis both for a specific territory and for a given period of time.

*4.2. The Architecture of the Proposed System*

The first stage in the implementation of the table semantic analysis algorithm is the design of the system architecture and its components (modules). It is necessary to design the data processing pipeline so that it consists of a sequence of replaceable modules. The advantage of this architecture is the support for easily adding new file types for processing. To do this, a parser must be written that will translate data from the new file format into an internal structure. At the design stage of the module, the following architecture was proposed, as shown in Figure 3.

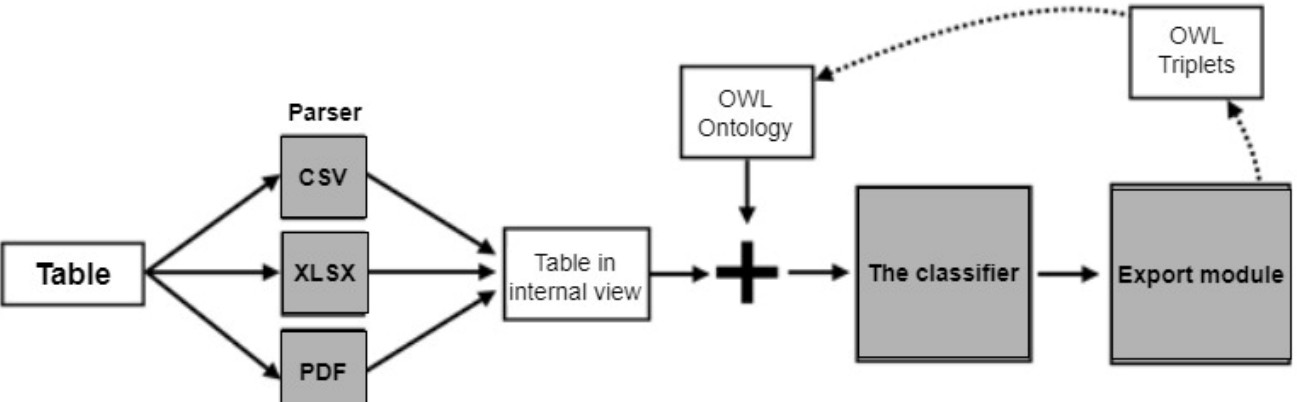

**Figure 3.** Software module architecture.

The input contains a table file (XLSX, CSV or PDF). Depending on the file type, the table is processed by the appropriate parser and translated into an internal structure for further processing. The internal structure of the table consists of two matrices—a matrix of types and a matrix of values (Figure 4). A value matrix is a direct mapping of table

cell values from a file into a matrix, a format that the algorithm can work with. The type matrix has the same dimensions as the value matrix but stores information about the cell's content type. When reading the table, the type matrix is filled with two types—"String" and "Number". The type "Number" is assigned to a cell if it contains a number or its value can be easily converted to a number (for example, the value "4254 km" can be converted to the number "4254" if the units are removed).

| Rivers_KZ | River_Length | River_Length_in_KZ |
|---|---|---|
| Irtysh | 4 254km | 1 700km |
| Ishim | 2 450 km | 1 400 km |
| Ural | 2 428 km | 1 082 km |
| Syrdarya | 2 219 km | 1 400 km |
| Tobol | 1 591 km | 800 km |
| Ile | 1 439 km | 815 km |
| Chu | 1 186 km | 800 km |
| Nura | 978 km | 978 km |
| Turgai | 825 km | 825 km |
| Oiyl | 800 km | 800 km |
| Sarysu | *800 km* | 800 km |
| Emba | 712 km | 712 km |
| Talas | 662 km | 453 km |
| Big Ozen | *650 km* | 260 km |
| Small Ozen | *638 km* | 395 km |
| Ilek | 623 km | 623 km |
| Irgiz | 593 km | 593 km |
| Sagiz | 511 km | 511 km |
| Shiderty | 502 km | 502 km |

(**a**)

| Rivers_KZ | River_Length | River_Length_in_KZ |
|---|---|---|
| Irtysh | Number | Number |
| Ishim | Number | Number |
| Text | Number | Number |
| Syrdarya | Number | Number |
| Tobol | Number | Number |
| Ile | Number | Number |
| Text | Number | Number |
| Nura | Number | Number |
| Turgai | Number | Number |
| Oiyl | Number | Number |
| Sarysu | Number | Number |
| Emba | Number | Number |
| Talas | Number | Number |
| Text | Number | Number |
| Text | Number | Number |
| Ilek | Number | Number |
| Irgiz | Number | Number |
| Sagiz | Number | Number |
| Shiderty | Number | Number |

(**b**)

**Figure 4.** (**a**) Type matrix before processing; (**b**) type matrix after processing at this stage.

After reading the table and the primary classification of cell types into the "String" and "Number" classes, the internal representation of the table enters the "Classifier" module, which also receives a file with an ontology in the OWL format. The ontology contains a description of the subject area—a set of objects, properties and relationships between them. This data is used in the process of table analysis to identify the dependence of relationships between cells and restore the semantic structure of the table.

In this module, the table is processed in several stages, and as a result, triplets are issued that were found during the semantic analysis of the table. These triplets are then added to the current ontology.

### 4.3. Ontology Loading Module

Before parsing the table, the OWL ontology is loaded using the Owlready2 library for Python.

Ontology analysis includes the following stages: analysis of objects, properties, relations and classes of objects.

During these stages, the lists and dictionaries listed below are completed:

object_names = []
data_properties = []
object_properties = []
class_names = []
name2object = {}
name2data_property = {}
name2object_property = {}

The object names are normalized and stored in object_names, and the corresponding object references are stored in name2object. These structures make it easier to parse the table. Subsequent modules use object_names to find objects and refer to objects to modify them in the ontology.

Similar actions are carried out for properties (data properties) and relations (object properties). For properties and relationships, accessing the object by reference is necessary to validate the property. For example, information can be retrieved about the domain and range of a property and find out if the property is correctly set in the parsed table. If the type of the value in the cell does not match the type of the value of the property, but a ghost can be made (for example, an integer in the cell, the property has the "real number" type), then the value is automatically converted to the required type. If type conversion is not possible (the cell contains a string that cannot be converted to a number, and the property is of type "integer"), a type conversion error will be generated.

The last step in ontology reading is class analysis. At this stage, a list of classes is compiled, which helps to analyze the table and identify the rows/columns containing objects.

After filling in the auxiliary data structures that facilitate the analysis of tables, the module proceeds to the next stage—reading the table from the file.

### 4.4. Tabular Data Reader Module

This module provides the first stage of table analysis. Depending on the file type (CSV, XLSX, PDF), the appropriate module is selected and the table is extracted from the file. If the input file has a CSV or XLSX extension, then it is enough to simply read the table from the file and remove, if any, empty rows and columns that do not carry any information. If processing a PDF file, a search must first be conducted for one or more tables in the document, extract them, and align columns and cells. After that, one can proceed to the next step.

After extracting the table, the module translates the table into a format that the program will work with in the future. The table in the internal representation of the program is stored in the form of two matrices—a matrix of values and a matrix of data types. In addition, at this stage, the primary classification of data into two types occurs—a string and a number. This will make data analysis easier in the future.

The type matrix at this stage can have four values: "Text", "Number", "Date," and an empty value. An empty value is set if the cell does not contain any information, or there are symbols indicating the absence of information, for example, a dash (hyphen, dash). The "Number" type is set if the cell contains a number, or its value can be easily reduced to a number (for example, by removing units of measurement).

### 4.5. Tabular Data Classification Module

After converting the table into an internal format, the module analyzes the contents of the cells and classifies the data into more specific types. To do this, several algorithms are used to determine whether the contents of a cell are a date, an object, or a property value.

The module also checks the table for errors and inconsistencies. The module finds duplicate rows and columns, the presence of empty rows or columns, cells with incorrect values, as well as the presence of cells with different data types in one row or column. If errors or inconsistencies are found in the table, the module issues an appropriate warning.

The module uses the Levenshtein distance [23] to search for objects in the ontology. Recall that the Levenshtein distance is a string metric for measuring the difference between two words (i.e., sequences of characters), informally defined as the minimum number of single-character edits (inserts, deletions or replacements of characters) required to replace one word with another. Thus, if there is a typo in the value in the table or there are additional symbols, the module will still be able to match the value of the table with an object or property in the ontology.

Additionally, the module can perform some additional data transformations. For example, a table may have cells with dates that need to be converted to a standard format

or cells with numerical values that need to be converted to a uniform order, such as thousands of kilometers to kilometers.

This module classifies data and analyzes the table structure. Using the structure of the knowledge area from the ontology, the module finds objects from the ontology, their attributes and the relationships between them. It specifies the current types of cells—instead of "Text," it can assign a unique object ID from the ontology if a match is found or determine which property the given cell reflects.

The first step in analyzing the structure of a table is row-by-row data processing. The module separately analyzes each row of the table and determines if the row contains a sequence of values that is a sequence of dates.

In addition, if there is a sequence of numbers in the table that is a sequence of years, those values are converted to dates. The same steps are repeated for each column.

The presence of a sequence of dates allows for the determination of the structure of the table and how it is further processed. So, if the table contains a sequence of dates, the module continues to analyze the table, taking into account the fact that rows (columns) contain objects, and each column (row) is a property of these objects for a specific date. For example, a table with population numbers (see Table 1) can have a similar structure where the rows contain regions, the columns list dates and the intersections of rows and columns contain numbers—the population in the given region from the row for the date indicated in the column. Typically, in such tables, the columns do not specify the type of the property, so a classifier must be applied to determine what the table describes.

If the table does not contain a sequence of dates, then the table is processed taking into account the fact that objects are located in the rows (columns) of the table, and their properties are listed in the columns (rows).

After determining whether there is a sequence of dates in the table and how to process it, it is necessary to analyze each cell and determine whether it contains an object, a property or a relation name or if it is a property or relation value.

If the cell contains a number, then it will be the value of the property. In addition, if the cell value is not a number, but this value can be converted to a number by removing the units of measurement, then when analyzing this cell, the units of measurement are removed, and the number, if necessary, is modified. For example, if a cell contains the value "1.46 million", it will be converted to "1,460,000".

If the cell contains text, then the closest name to the cell value is sequentially searched using the Levenshtein distance in the object_names, data_properties, object_properties, and class_names lists that were filled during ontology loading. This match of two words is correct if the distance between the found name and the text from the table cell does not exceed 30% of the length of the text in the cell. In other words, to get the found word from the cell value, the number of character replacement/insertion/deletion operations should not exceed 30% of the original word. This threshold is set so that the search can match misspelled words with names from the ontology, but at the same time, incorrect concepts are not matched. Another reason why the threshold is set to 30% is the fact that sometimes, the names of properties indicate units of measurement. Thus, when searching using the Levenshtein distance, a threshold of 30% in most cases will allow for the removal of units of measurement from the cell value and comparison with a property from the ontology.

Processed tables have an enumeration of objects in rows and columns, an enumeration of their relations or properties. There may also be an opposite structure—in the rows, the enumeration of relations or properties, and in the columns, the enumeration of objects. To take into account the features of these structures, during the analysis of the cells, auxiliary data structures are also filled in, which will be further used when exporting data:

1. row2object
2. row2predicate
3. row2date
4. col2object
5. col2predicate

6. col2date

These structures describe which object, predicate (in this case, the predicate combines properties and relations), or date refers to the row/column number of the cell being processed. If a term from the ontology is found in a cell, then the corresponding row/column dictionary is entered in a pair (Row/column index, found term—object, predicate or date). These structures are always filled only in the first row or in the first column of the table since it is in the first rows/columns that the content type of the entire column or row is indicated, and in the subsequent rows/columns, the predicate values are listed.

The result of the table analysis at this stage is a filled matrix of cell types. It reflects the found objects from the ontology, properties, relationships and their values in text or numerical form. In other words, this type of matrix is the semantic markup of a table that describes the semantic meaning of each cell (see Figure 4).

*4.6. Data Export Module*

After receiving the semantic markup of the table at the previous stage, it is necessary to export the received information. The data export module is responsible for this. It takes as input a table with cell values, a semantic markup of the table, and auxiliary data structures that were filled in at the ontology loading stage (lists of classes, objects, properties, relations, and dictionaries for converting a name into an object reference, a property, and a relation).

The module goes through each row and each column sequentially. For each cell, one can define col_idx and row_idx—column index and row index, respectively. When processing a cell, two conditions must be met for a successful export:

1. The index of the column in which the col_idx cell is located must be present in exactly one of the following dictionaries: *col2object*, *col2predicate*, and *col2date*

2. The row index in which the row_idx cell is located must be present in exactly one of the following dictionaries: *row2object, row2predicate*, and *row2date.*

The fulfillment of these conditions allows for the determination of the object and predicate to which the value of this cell refers. For example, if a row contains an object and a column contains a property, then the numeric value at the intersection of the row and column will be the property of that object.

Once the object and predicate are defined, the cell value must be checked to determine if it is of the appropriate type. To do this, using col2predicate or row2predicate (depending on the structure of the table), the name of the predicate is taken, then using name2data_property or name2object_property, a reference to the property or relation object is determined. An object describing a property or relation has detailed information about the predicate, such as domain, range, whether the predicate is functional, inverse, and so on.

With the help of an object with a description of the property, a number of checks are carried out:

1. Whether the class of the object on which the property will be set belongs to the domain of the property.

2. Whether the cell value belongs to the range of the predicate (or is it possible to cast the cell value to the required type, for example, convert an integer to a real number). If the predicate is a property, then the cell value must contain a string or a number. If the predicate is a relation, then the cell must contain a subject or a list of subjects.

3. Whether the given predicate has an inverse predicate. This condition is relevant only for relations between objects. If the predicate has a predicate inverse to it, then when adding a property to the row/column object, it will be necessary to add the inverse property to the object from the table cell (for example, the lake is in the region, then the region contains the lake).

4. The functionality of the predicate is checked. This condition does not allow more than one property of a given type to be added to an object (for example, a person can have only one age) if this predicate is specified as a functional predicate.

If all conditions are met, then the triplet (object, predicate, cell value) is written into the ontology. Moreover, if the predicate is a relationship between objects, then the cell value can contain many objects (for example, a region contains a list of cities). In this case, a triplet must be added for each subject from the cell.

As a result of the operation of the export module, the resulting table markup, together with the table data, is converted into triplets, which are then added to the ontology. The result of the work is shown in Figure 5.

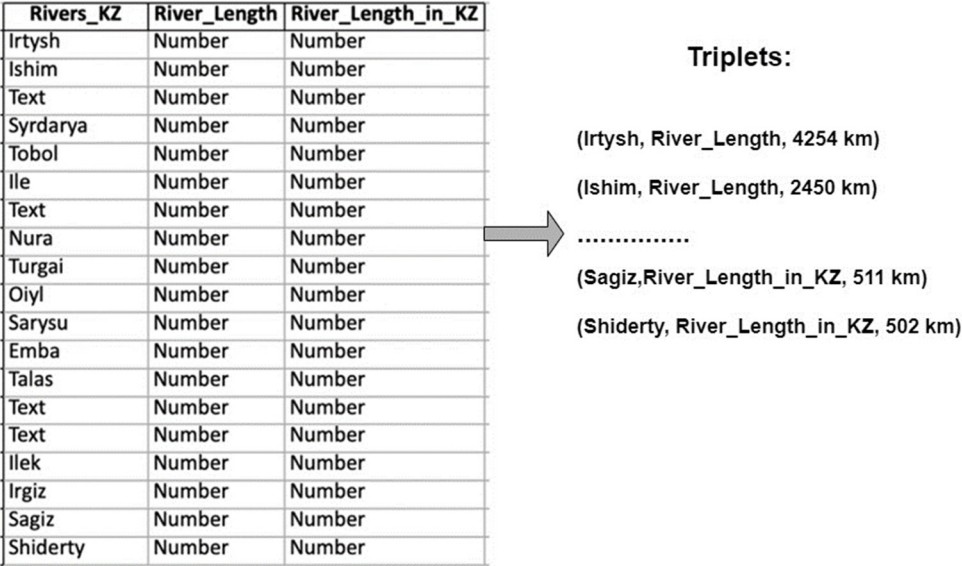

**Figure 5.** Translation of analysis results into triplets.

After completing the process of filling the OWL ontology, one can create queries that retrieve related data about water resources and regions; below are examples of useful queries:

1. Query 1 implements the derivation of water objects, the regions to which these objects belong and the population of this region in the time interval.
2. Query 2 displays water bodies and their WPI indicators and their pollution class based on WPI indicators.
3. Query 3 displays water bodies with the given WPI indicators.
4. Query 4 displays water bodies with high WPI values and the number of people suffering from diseases of the circulatory system associated with iodine deficiency. The query result is shown in Figure 6.

Figure 7 shows a visualization of Request 4, which shows the work of the developed application. This request allows for the data on water bodies, regions, and populations to be obtained. All these data were obtained automatically from different tables in the process of performing semantic analysis. It is possible to analyze data by linking them together by coordinates in space or by belonging to time intervals.

This ontology provides new opportunities for studying information about the water resources of Kazakhstan, facilitating the process of analyzing data obtained from various sources and combined in one place.

Thus, an ontology was developed that describes the area of knowledge, an analysis of existing solutions was carried out, and an own module was developed, taking into account the shortcomings of other solutions. As a result, a universal module was obtained that can analyze data in various fields of knowledge, provided that an ontology has been developed for this area.

| All | All | All | All |
|---|---|---|---|
| Bolshaya_Almatinka | 1.34 | Almaty_City | 4017.8 |
| Big_Chebache | 0.66 | Akmola_Region | 3647.9 |
| Borovoe | 0.52 | Akmola_Region | 3647.9 |
| Ishim | 1.11 | Akmola_Region | 3647.9 |
| Nura | 1.17 | Akmola_Region | 3647.9 |
| Shuchie | 0.44 | Akmola_Region | 3647.9 |
| Vyacheslavskoe | 0.67 | Akmola_Region | 3647.9 |
| Assa | 1.47 | Zhambyl_Region | 3623.2 |
| Shu | 2.84 | Zhambyl_Region | 3623.2 |
| Talas | 0.92 | Zhambyl_Region | 3623.2 |
| Bukhtarma | 3.22 | East_Kazakhstan_Region | 3231.7 |
| Irtysh | 0.81 | East_Kazakhstan_Region | 3231.7 |
| Uba | 3.45 | East_Kazakhstan_Region | 3231.7 |
| Ulba | 4.76 | East_Kazakhstan_Region | 3231.7 |

**Figure 6.** Implementation of Query4 in the developed application.

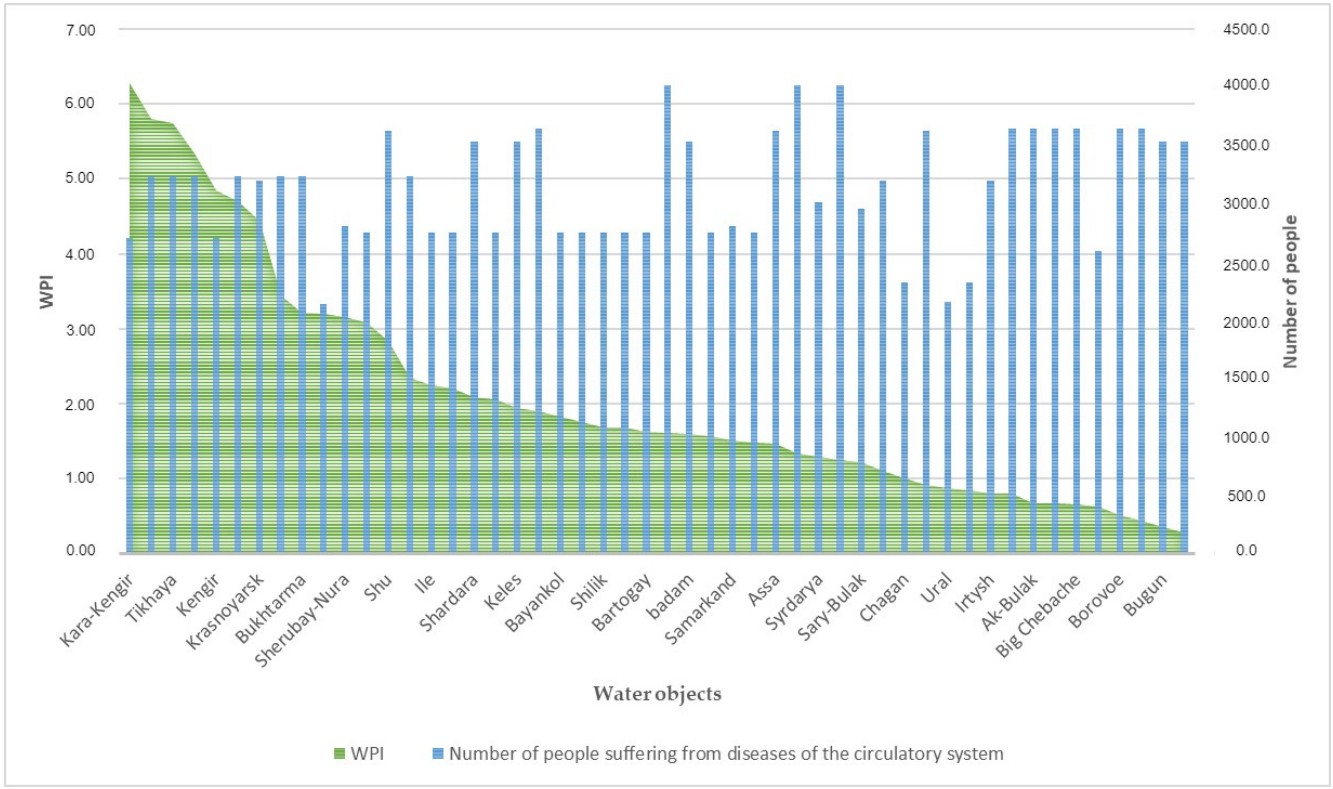

**Figure 7.** Visualization of Query 4.

The developed solution will significantly speed up the analysis of tabular data and allow the analysis of semantically related data, which will provide an opportunity to get a more complete picture of the field of research.

## 5. Results

### 5.1. Experimental Setup

Before starting testing, it is necessary to determine the requirements that the table semantic analysis algorithm must satisfy.

The developed algorithm must extract at least 95% of the triplets from the table. To successfully extract a triple, the algorithm must recognize three elements for each triple—subject, predicate and object. In addition, the accuracy of determining each individual

subject, predicate and object must exceed 98%. Accuracy is understood as the correctness of the correlation of a table value with an object from the ontology—the selected object from the ontology must be correct in meaning. To assess the quality of the algorithm, only a manual assessment method seems possible since automatic assessment requires a pre-marked dataset, which was not found for the task of semantic analysis of tables in the field of natural and economic resources.

For testing, 30 tables in the field of natural and economic resources were selected. The tables contain information on various indicators of the regions, for example, population statistics, the number of male/female, urban/rural population, there is information on the economic well-being of residents. There are also tables that describe water resources: rivers, lakes, and reservoirs. There are data on depth, length, and average daily water flow, i.e., the amount of water that passes through a certain section of the river per day.

The algorithm was run on each of the selected tables, and the resulting triplets were exported to the OWL ontology. An example of a table is shown in Figure 8. In addition, to calculate statistics for each table, its markup was saved. An example of a markup file is shown in Figure 9.

| Regions_KZ | 2000 | 2001 | 2002 | 2003 | 2004 | ... | 2021 |
|---|---|---|---|---|---|---|---|
| Kazakhstan | 14,901,641 | 14,865,610 | 14,851,059 | 14,866,837 | 14,951,200 | ... | 18,879,552 |
| Akmola_region | 799,179 | 776,377 | 755,000 | 748,167 | 748,930 | ... | 735,566 |
| Aktobe_region | 677,715 | 670,231 | 668,166 | 668,378 | 671,812 | ... | 894,333 |
| Almaty_region | 1,557,141 | 1,554,320 | 1,554,573 | 1,560,267 | 1,571,194 | ... | 2,077,967 |
| Atyrau_region | 441,692 | 443,630 | 447,634 | 451,928 | 457,215 | ... | 657,110 |
| West_Kazakhstan_region | 609,161 | 601,648 | 600,330 | 602,133 | 603,832 | ... | 661,316 |
| Zhambyl_region | 986,144 | 981,928 | 979,199 | 980,072 | 985,552 | ... | 1,139,192 |
| Karaganda_region | 1,390,454 | 1,364,781 | 1,344,244 | 1,333,656 | 1,330,927 | ... | 1,375,938 |
| Kostanay_region | 988,787 | 959,274 | 935,717 | 919,558 | 913,435 | ... | 864,550 |
| Kyzylorda_region | 598,526 | 599,738 | 600,972 | 603,804 | 607,491 | ... | 814,588 |
| Mangistau_region | 315,203 | 319,191 | 328,265 | 338,612 | 349,668 | ... | 719,571 |
| Pavlodar_region | 790,774 | 772,453 | 758,154 | 748,651 | 745,238 | ... | 751,012 |
| North_Kazakhstan_region | 713,628 | 702,623 | 691,263 | 682,148 | 674,497 | ... | 543,735 |
| Turkestan_region | 2,005,023 | 2,051,351 | 2,079,506 | 2,111,893 | 2,150,256 | ... | 2,044,742 |
| East_Kazakhstan_region | 1,516,785 | 1,499,097 | 1,482,550 | 1,465,931 | 1,455,412 | ... | 1,363,797 |
| Astana | 380,990 | 440,209 | 493,062 | 501,998 | 510,533 | ... | 1,184,411 |
| Almaty | 1,130,439 | 1,128,759 | 1,132,424 | 1,149,641 | 1,175,208 | ... | 1,977,258 |
| Shymkent | - | - | - | - | - | ... | 1,074,466 |

**Figure 8.** An example of a test table.

| Class_Regions_KZ | Year_2000 | Year_2001 | ... | Year_2021 |
|---|---|---|---|---|
| Kazakhstan | 2000_14901641.0 | 2001_14865610.0 | ... | 2021_18879552.0 |
| Akmola_Region | 2000_799179.0 | 2001_776377.0 | ... | 2021_735566.0 |
| Aktobe_Region | 2000_677715.0 | 2001_670231.0 | ... | 2021_894333.0 |
| Almaty_Region | 2000_1557141.0 | 2001_1554320.0 | ... | 2021_2077967.0 |
| Atyrau_Region | 2000_441692.0 | 2001_443630.0 | ... | 2021_657110.0 |
| West_Kazakhstan_Region | 2000_609161.0 | 2001_601648.0 | ... | 2021_661316.0 |
| Zhambyl_Region | 2000_986144.0 | 2001_981928.0 | ... | 2021_1139192.0 |
| Karagandy_Region | 2000_1390454.0 | 2001_1364781.0 | ... | 2021_1375938.0 |
| Kostanay_Region | 2000_988787.0 | 2001_959274.0 | ... | 2021_864550.0 |
| Kyzylorda_Region | 2000_598526.0 | 2001_599738.0 | ... | 2021_814588.0 |
| Mangystau_Region | 2000_315203.0 | 2001_319191.0 | ... | 2021_719571.0 |
| Pavlodar_Region | 2000_790774.0 | 2001_772453.0 | ... | 2021_751012.0 |
| North_Kazakhstan_Region | 2000_713628.0 | 2001_702623.0 | ... | 2021_543735.0 |
| Turkestan_Region | 2000_2005023.0 | 2001_2051351.0 | ... | 2021_2044742.0 |
| East_Kazakhstan_Region | 2000_1516785.0 | 2001_1499097.0 | ... | 2021_1363797.0 |
| Astana | 2000_380990.0 | 2001_440209.0 | ... | 2021_1184411.0 |
| Almaty | 2000_1130439.0 | 2001_1128759.0 | ... | 2021_1977258.0 |

**Figure 9.** An example of the resulting markup.

Each cell contains an indication of a specific entity from the ontology with which it was associated. The first column lists the regions that have been linked to the cells. The remaining columns contain converted values prefixed with the year they refer to. Similar markup files were obtained for each table. Then, they were manually checked, and errors in the work of the classifier were noted.

### 5.2. Evaluation

The quality of the algorithm was evaluated by accuracy, precision, recall, as well as the F1-measure metric, which considers both metrics. The calculation of metrics is carried out separately for each component of the knowledge triplets—objects, predicates, and subjects. Therefore, we will further understand further an element from the union of sets of objects, predicates, and subjects.

For each class, we calculate the following parameters:

1. True Positive (TP)—correct definition.
2. True Negative (TN)—correct definition.
3. False Positive (FP)—type I error.
4. False Negative (FN)—type II error.

Based on the obtained parameters, we calculate the following quality metrics:

1. Accuracy is a metric used to measure the performance of a classification model.
2. Precision is the proportion of objects that the algorithm correctly identified relative to all objects that the algorithm identified.
3. Recall is the proportion of objects that the algorithm correctly identified relative to all objects that are positive.
4. F1-score is the harmonic mean between precision and recall. The F1 score shows the balance between precision and recall.

Appendix A shows Table A1, which consists of predicates and their metric indicators.

For example, for predicates, a total of 8710 predicates were detected, of which 8691 were correctly detected, and 8750 were all predicates. Thus, Formulas (1)–(4) demonstrate the calculation of metrics:

$$\text{Accuracy} = \frac{\text{TP}}{\text{TP} + \text{FN} + \text{FP}} = \frac{8691}{8750 + 19} = 0.9911 \tag{1}$$

$$\text{Precision} = \frac{\text{TP}}{\text{TP} + \text{FP}} = \frac{8691}{8710} = 0.9978 \tag{2}$$

$$\text{Recall} = \frac{\text{TP}}{\text{TP} + \text{FN}} = \frac{8691}{8750} = 0.9932 \tag{3}$$

$$\text{F1} = \frac{2 * \text{Precision} * \text{Recall}}{\text{Precision} + \text{Recall}} = \frac{2 * 0.9978 * 0.9932}{0.9978 + 0.9932} = 0.9955 \tag{4}$$

After the calculations, the following metrics for subjects, predicates and objects were obtained, as well as the final quality metrics for the classification of entities (Table 2).

**Table 2.** Metrics for subjects, predicates, and objects, as well as final quality metrics for entity classification.

| Metrics | Discovered Subjects (TP + FP) | Correct Subjects (TP) | Total Subjects (TP + FN) | Macro Precision | Macro Recall | Macro F1 | Accuracy |
|---|---|---|---|---|---|---|---|
| Subject classification | 8724 | 8709 | 8750 | 99.83% | 99.53% | 99.68% | 99.36% |
| Predicate classification | 8710 | 8691 | 8750 | 99.78% | 99.33% | 99.55% | 99.11% |
| Object classification | 8746 | 8743 | 8750 | 99.97% | 99.92% | 99.94% | 99.88% |
| Final metrics | 26,180 | 26,143 | 26,250 | 99.86% | 99.59% | 99.72% | 99.45% |

As can be seen from Table 2, the algorithm has high accuracy, recall, and F1 measures. Misprints, use of synonyms, or other inaccuracies lead to a decrease in completeness, but adding varieties of spelling of words to the ontology allows for this metric to be increased.

In the process of testing, 8697 triples were extracted and exported into the ontology, in which the subject, predicate and object were simultaneously defined. Considering that there were 8750 triplets in the tables, the completeness of extracting triplets is 99.4%, which significantly exceeds the threshold of 95% defined in the testing requirements.

Such high metrics were obtained by using the Levenshtein distance to match entities and using ontology as a source of information about the structure of the knowledge area. This approach makes it possible to achieve a good quality object classification without the need to make changes to the algorithm when adding new entities or changing the knowledge area.

### 5.3. Comparative Analysis with Similar Algorithms

Below, in Table 3, there is a comparative analysis between our proposed solution and alternative approaches.

**Table 3.** List of properties of similar algorithms with our solution.

| Algorithms | Iterative Approach | Source Types | Custom Data Format Support | Versatility across Knowledge Domains | Time Efficiency | OWL-Compatible Data Exportation |
|---|---|---|---|---|---|---|
| Our solution | Yes | Excel | Yes | Yes | No | Yes |
| TableMiner+ | Yes | HTML, PDFs, XML | No | No | Yes | No |
| QTLTableMiner++ | No | HTML, PDFs, XML | No | No | Yes | Yes |
| Qurma | No | HTML, PDFs | Yes | Yes | No | Yes |
| Camelot | No | PDFs | Yes | Yes | No | No |

Our method stands out in its comprehensive approach to table interpretation. Using an iterative algorithm emphasizes accuracy and adaptability across diverse data structures and knowledge fields. Although not the swiftest, it strikes a balance between speed and capability, and uniquely, it exports data to OWL for integration with ontology-based systems.

While processing speed in semantic table interpretation has often been prioritized, its importance needs reevaluation. In specific applications, accuracy and precision may outweigh speed. This is especially true for critical scientific research or decisions where the utmost accuracy in tabular data interpretation is essential, even at the cost of longer processing times.

Furthermore, if an algorithm's computational effort is mainly expended during its initial indexing phase, the subsequent interactions could be much faster. This makes the initial longer processing time a worthwhile investment. It's also possible that decisions were made in algorithm design to favor aspects like simplicity and resource efficiency over sheer speed.

## 6. Discussion

The iterative algorithm developed in this study represents a significant advance in the field of table semantic analysis. Using an ontology-based approach, the algorithm can perform an initial analysis of the table, determining the content type of each cell and then refine its type based on the ontology. This method ensures accurate data extraction and facilitates the addition of new data to the ontology, improving the analysis of subsequent tables.

The current approach illustrates a novel methodology for table semantic analysis through an iterative algorithm. Unlike conventional methods, this approach harnesses the power of ontology-based analysis, allowing for a more intricate, two-tiered procedure.

In previous works, semantic analysis of tables often involved singular, straightforward analyses, which occasionally suffered from inaccuracies due to the complexities and variability of tabular data [24]. However, the newly developed algorithm initiates with an overarching examination of the table, wherein it discerns the content type of each cell. This

primary step already surpasses many traditional methods by creating a preliminary yet solid framework for the next steps [25].

Furthermore, not only does this iterative algorithm ensure accurate data extraction, but it also aids in adding new data seamlessly into the ontology. This, in turn, enriches the ontology, which improves the analysis of tables in subsequent runs. It's a feedback loop that consistently improves upon itself [26]. Traditional methods, on the other hand, lacked this continuous improvement approach, which often made them stagnate in their accuracy levels over time [27].

In conclusion, the iterative algorithm is not just a minor upgrade; it's a significant advancement in the realm of table semantic analysis. By incorporating an ontology-based approach and refining data extraction through a two-tiered analysis, it addresses the pitfalls of previous works, offering a dynamic, self-improving solution for the challenges of tabular data extraction.

The innovation of this research lies in its automated method of extracting data from tables in different languages and in different fields of knowledge, provided that an appropriate ontology is available. The integration of advanced techniques such as cosine distance search and neural network-based table subject classification further highlights its effectiveness. The result of this work was a software application capable of semantically classifying tabular data. Taking a table and an ontology as input, it outputs the identified table objects and their attributes based on the knowledge structure of the ontology. This achievement paves the way for a faster transition of information from tables to ontologies, making it easier to summarize data and identify patterns.

A rigorous testing phase, including 30 natural and economic resource tables, confirms the robustness and accuracy of the algorithm. Evaluation metrics, including precision, recall, and F1-score, provide a comprehensive assessment of the algorithm's performance. The results presented in Tables 1–3 demonstrate the high accuracy, completeness, and F1 measures of the algorithm. Despite typos, synonyms, or other inaccuracies that may lead to a slight decrease in completeness, including different spellings of words in the ontology can improve this indicator. During testing, 8697 triples were extracted and integrated into an ontology in which the subject, predicate and object are simultaneously defined. Considering that the tables contained 8750 triples, the completeness of triple extraction is an impressive 99.4%, well above the 95% threshold set in the testing requirements.

Such notable metrics have been achieved through the use of Levenshtein distance for matching entities and the use of ontology as a source of information about the structure of the knowledge area. This approach allows for achieving a high-quality object classification without the need to make changes to the algorithm when adding new entities or changing the knowledge area.

However, like any innovative solution, continuous improvement and testing in various scenarios will only improve its capabilities. It would be interesting to know how the algorithm scales with large datasets and how it performs across different domains. The potential for integrating this algorithm with other systems, such as databases, content management systems, or even web crawlers, can be explored. Implementing a feedback mechanism for users can help further improve the algorithm.

While our results have been promising, especially in the field of water resources and socio-economic indicators of Kazakhstan, we do recognize that every algorithm has its boundaries.

The following are the limitations:

1. Data Dependency on Ontologies: Our method's reliance on ontologies is both its strength and limitation. The quality and comprehensiveness of the ontology directly affect the performance of the algorithm. The absence of a robust ontology for a particular field might limit the adaptability of our approach in such contexts.
2. Complexity of Table Structures: Though our algorithm successfully parsed varied table structures during our tests, there may be instances of highly irregular or complex tables that could challenge the algorithm's performance.

3. Scalability: While our tests included 30 tables, larger-scale extraction tasks, especially those involving thousands of tables, have not been rigorously tested.
4. Language Generalization: Even though our method can handle tables in any language, certain linguistic nuances or rare terminologies might affect the precision of data extraction in lesser-known languages.

The future perspectives and research lines arising from this work are manifold:

1. Algorithm Enhancement for Complex Tables: One potential research avenue is refining our iterative algorithm to handle even more complex and irregular table structures.
2. Integration with Machine Learning Models: There's a promising direction in integrating state-of-the-art machine learning models to improve data classification further, especially in recognizing linguistic nuances across various languages.
3. Scalability Tests: We envision extensive scalability tests to ascertain the algorithm's performance on larger datasets.
4. Expanding Ontology Repositories: Given our method's reliance on ontologies, there's a valuable line of research in developing comprehensive ontologies across various fields. This will benefit not only our method but also other ontology-reliant technologies.
5. Feedback Mechanisms: Implementing a feedback mechanism where the software application learns from any inaccuracies and refines its data extraction techniques in real time could be another exciting avenue to explore.
6. Collaborative Features: Considering the practical application of our software, introducing collaborative features, allowing teams to work simultaneously and make annotations, would make it even more useful for larger projects.

In conclusion, the iterative algorithm developed in this study has significant potential in the field of semantic analysis of tables. Its modular approach, combined with cutting-edge technology, makes it stand out from the crowd. The high accuracy and completeness achieved during the test phase highlight its potential as a robust and versatile tool for semantic analysis of tables in various domains.

## 7. Conclusions

The main scientific significance and novelty of our research lies in solving the problems of semantic analysis of tables using an innovative iterative algorithm, offering a comprehensive solution based on ontology. This is critical given the ubiquity of data in tabular format across all sectors. The practical implications of the research are enormous, as evidenced by the high accuracy of the algorithm in tests, indicating its potential usefulness in various applications.

This study delved into the intricacies of table data extraction and semantic analysis, emphasizing the importance of understanding the semantics of tables in order to extract meaningful relationships and information. The main goal was to develop a technology for the semantic analysis of tables using the OWL ontology, a recognized formal knowledge description language.

The proposed iterative algorithm is distinguished by its ability to perform an initial analysis of the table, determining the content type of each cell and then refining its type based on the ontology. This method ensures accurate data extraction and facilitates the addition of new data to the ontology, improving the analysis of subsequent tables.

The real-world implications of our research are vast. With our method, researchers, data analysts, and industry professionals can efficiently extract and interpret data from tables, thereby streamlining data integration, analysis, and decision-making processes. For instance, our rigorous testing phase demonstrated an impressive triple extraction recall of 99.4% from tables related to water resources and socio-economic indicators of Kazakhstan. Such high accuracy indicates the algorithm's potential in diverse applications, ranging from environmental studies to socio-economic analyses.

The result of this effort was a software application capable of semantically classifying tabular data. Taking a table and an ontology as input, it outputs the identified table objects and their attributes based on the knowledge structure of the ontology. This achievement

paves the way for a faster transition of information from tables to ontologies, making it easier to summarize data and identify patterns.

A rigorous testing phase, including 30 tables in the field of natural and economic resources, confirms the reliability and accuracy of the algorithm. Evaluation metrics, including precision, recall, and F1 score, provide a comprehensive assessment of the algorithm's performance.

The results presented in Tables 1–3 demonstrate the high accuracy, completeness, and F1 measures of the algorithm. Despite typos, synonyms, or other inaccuracies that may lead to a slight decrease in completeness, including different spellings of words in the ontology can improve this indicator. In the process of testing, 8697 triplets were extracted and integrated into an ontology in which the subject, predicate and object are simultaneously defined. Given that the tables contained 8750 triplets, the triplet extraction completeness is an impressive 99.4%, well above the 95% threshold set in the testing requirements.

Such noteworthy metrics have been achieved through the use of Levenshtein distance to match entities and the use of ontology as a source of information about the structure of a knowledge domain. This approach allows for high-quality object classification to be achieved without the need to make changes to the algorithm when adding new entities or changing the knowledge area.

**Supplementary Materials:** The following supporting information can be downloaded at https://github.com/Titrom025/PyTableMiner/blob/main/webApp/templates/index.html.

**Author Contributions:** Conceptualization, M.M. and V.B.; methodology, V.B.; software, R.T.; resources and data curation, A.O.; writing—original draft preparation, V.B. and A.O.; writing—review and editing, M.M.; visualization, R.T.; supervision, M.M. All authors have read and agreed to the published version of the manuscript.

**Funding:** This research was funded by the Ministry of Science and Higher Education of the Republic of Kazakhstan: AP09261344 "Development of methods for automatic extraction of spatial objects from heterogeneous sources for information support of geographic information systems".

**Institutional Review Board Statement:** Not applicable.

**Informed Consent Statement:** Not applicable.

**Data Availability Statement:** All tables utilized in this study are available in the "Tables" folder of our GitHub repository, which can be accessed via the following link: https://github.com/Titrom025/PyTableMiner/tree/main/Tables. Additionally, the updated ontology relevant to our research is stored in the "ontologies" directory of the same repository. Direct access to the ontology data is available at: https://github.com/Titrom025/PyTableMiner/tree/main/ontology. For those interested in a broader overview of the project's structure and components, the main web application template can be reviewed at Supplementary Materials.

**Conflicts of Interest:** The authors declare no conflict of interest.

## Appendix A

**Table A1.** Predicate classification metrics.

| Predicates | Detected (TP + FP) | Correctly Detected (TP) | Total Predicates (TP + FN) | Precision | Recall | F1 | Accuracy |
|---|---|---|---|---|---|---|---|
| Average_annual_water_consumption | 21 | 21 | 21 | 100.00% | 100.00% | 100.00% | 100.00% |
| Basin_has_River | 17 | 17 | 18 | 100.00% | 94.44% | 97.14% | 94.44% |
| Basin_Square | 8 | 8 | 8 | 100.00% | 100.00% | 100.00% | 100.00% |
| Basins_Population | 8 | 8 | 8 | 100.00% | 100.00% | 100.00% | 100.00% |
| Contains | 95 | 95 | 98 | 100.00% | 96.94% | 98.45% | 96.94% |
| Female_Population | 379 | 379 | 379 | 100.00% | 100.00% | 100.00% | 100.00% |
| Lake_Square | 67 | 66 | 67 | 98.51% | 98.51% | 98.51% | 97.06% |
| Located_in | 95 | 95 | 98 | 100.00% | 96.94% | 98.45% | 96.94% |
| Male_Population | 379 | 379 | 379 | 100.00% | 100.00% | 100.00% | 100.00% |
| Population | 379 | 379 | 379 | 100.00% | 100.00% | 100.00% | 100.00% |
| River_Fall | 21 | 21 | 21 | 100.00% | 100.00% | 100.00% | 100.00% |
| River_in_Basin | 17 | 17 | 18 | 100.00% | 94.44% | 97.14% | 94.44% |

**Table A1.** *Cont.*

| Predicates | Detected (TP + FP) | Correctly Detected (TP) | Total Predicates (TP + FN) | Precision | Recall | F1 | Accuracy |
|---|---|---|---|---|---|---|---|
| River_Length | 19 | 19 | 19 | 100.00% | 100.00% | 100.00% | 100.00% |
| River_Length_in_Basin | 8 | 8 | 8 | 100.00% | 100.00% | 100.00% | 100.00% |
| River_Length_in_KZ | 40 | 40 | 40 | 100.00% | 100.00% | 100.00% | 100.00% |
| Rural_Female_Population | 330 | 330 | 330 | 100.00% | 100.00% | 100.00% | 100.00% |
| Rural_Male_Population | 330 | 330 | 330 | 100.00% | 100.00% | 100.00% | 100.00% |
| Rural_Population | 330 | 330 | 330 | 100.00% | 100.00% | 100.00% | 100.00% |
| Urban_Female_Population | 379 | 379 | 379 | 100.00% | 100.00% | 100.00% | 100.00% |
| Urban_Male_Population | 379 | 379 | 379 | 100.00% | 100.00% | 100.00% | 100.00% |
| Urban_Population | 379 | 379 | 379 | 100.00% | 100.00% | 100.00% | 100.00% |
| Water_and_energy_resources_Energy | 21 | 19 | 22 | 90.48% | 86.36% | 88.37% | 79.17% |
| Water_and_energy_resources_Power | 11 | 10 | 11 | 90.91% | 90.91% | 90.91% | 83.33% |
| Water_Consumption | 2909 | 2903 | 2923 | 99.79% | 99.32% | 99.55% | 99.00% |
| Water_Level | 2081 | 2072 | 2098 | 99.57% | 98.76% | 99.16% | 98.00% |
| Water_resources_KZ | 8 | 8 | 8 | 100.00% | 100.00% | 100.00% | 100.00% |
| **Total** | **Detected Predicates** 8710 | **Correctly Detected** 8691 | **Total Predicates** 8750 | **Macro Precision** 99.78% | **Macro Recall** 99.33% | **Macro F1** 99.55% | **Accuracy** 97.67% |

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
