# Peer review of "Ontology-Driven Semantic Analysis of Tabular Data: An Iterative Approach with Advanced Entity Recognition"

_applsci, doi:10.3390/app131910918_

Round 1

Reviewer 1 Report

1. "For the developed algorithm, several rules were identified that help determine the 290 structure of the table" - Can the authors explain on what basis the rules are determined?

2. Figure 2 is not precise, so I can't understand the developed ontology.

3.  I strongly suggest to replace all the figures with good quality.  From the images in the article, I can't understand what the authors are trying to convey.

4. There is no comparison with similar algorithms. I suggest including the previous work comparison table.

Author Response

Dear Reviewer,

We would like to express our sincere gratitude for taking the time to review our manuscript and for your constructive comments and suggestions. Your feedback has been invaluable in improving the quality of our work. 

Our answers are given in the uploaded file, please check the attached file.

Reviewer 2 Report

The authors have undertaken a noteworthy endeavor in proposing an iterative algorithm that leverages ontology for semantic analysis, offering promising prospects across diverse linguistic and multidisciplinary contexts. Furthermore, the integration of advanced techniques such as cosine distance search and neural network-based table subject classification underscores their commitment to enhancing the efficiency of their proposed methodology. Additionally, the authors' rigorous testing, encompassing a dataset related to water resources and socio-economic indicators of Kazakhstan, adds substantial credibility to their research. The remarkable triple extraction recall rate of 99.4% attests to the effectiveness of their approach, with the utilization of Levenshtein distance and ontology as key sources of information contributing significantly to achieving these impressive metrics. While the authors have made substantial contributions, there are several areas within the manuscript that require attention and revision to further elevate its quality and impact.

Performance Comparison Table: It is recommended that the authors include a performance comparison table in the article, comparing their method with other relevant algorithms. This can help readers gain a clearer understanding of the performance of the authors' method in various aspects and provide more information about the advantages of their method.

Image Quality: Point out the issue of image quality in the article and suggest that the authors recreate these images to enhance their professionalism. Screenshots are generally not suitable for academic articles, and authors should use professional graphics tools to create high-quality charts and graphics.

Explanation of Scientific Significance: It is recommended that the authors provide a more detailed explanation of the scientific significance of the various data they have analyzed. Explain the importance of this data within the field and how their method can be applied to address practical issues. This will help readers better understand the research's background and contributions.

Font Size in Flowcharts: Highlight the issue of small font size in the relevant algorithm flowcharts, making them difficult to read, and suggest that the authors increase the font size to improve readability. Ensure that the flowcharts can clearly convey the steps and processes of the method.

Image Clarity: Mention the problem of blurriness in Figure 7 and recommend that the authors regenerate or replace this figure to ensure its clarity. Blurry images can potentially affect readers' understanding of the research results.

There is some minor errors on the English presentation. Overall, it's acceptable.

Author Response

(The authors gave the same response as above.)

Reviewer 3 Report

This paper is well written, complete in structure and correct in derivation, and I think it can be accepted.

Author Response

Dear Reviewer,

Thank you for your thoughtful review and positive feedback on our paper. We are truly encouraged by your appreciation of the structure, clarity, and correctness of our work.

Your recognition further motivates us to maintain and uphold these standards in our future endeavors. The acceptance and trust you have shown in our research are deeply appreciated and reinforces our commitment to producing quality research for the benefit of the scientific community.

We are grateful for your time and effort in evaluating our manuscript, and we value the professionalism and thoroughness you brought to the review process.

Once again, thank you for your valuable feedback and kind words. We remain open to any further suggestions or insights you may have now or in the future.

Warm regards,

Madina Mansurova

Reviewer 4 Report

This is study is original and clearly presents the main idea, methods, literature, difficulties and solutions about the proposed problems. 

The authors study the extraction and semantic analysis of data from tables, emphasizing the importance of understanding the semantics of tables to obtain useful information. The main aim is to construct a technology using the ontology for the semantic analysis of tables. An iterative algorithm has been proposed that can parse the contents of a table and determine cell types based on the ontology. The study presents an automated method for extracting data in various languages, in various fields, subject to the availability of an appropriate ontology. Advanced techniques such as cosine distance search and table subject classification based on a neural network have been integrated to increase efficiency. The result is a software application capable of semantically classifying tabular data, facilitating the rapid transition of information from tables to ontologies. Rigorous testing, including tables in the field of water resources and socio-economic indicators of Kazakhstan, confirmed the reliability of the algorithm. The results demonstrate high accuracy with a notable triple extraction recall of 99.4%. The use of Levenshtein distance for matching entities and ontology as a source of information was key to achieving these metrics. The study offers a promising tool for efficiently extracting data from tables.

This paper can be accepted without any revision.

Author Response

Dear Reviewer,

We are deeply grateful for your comprehensive review and positive evaluation of our study. Your acknowledgment of the originality and clarity of our research, as well as the detailed understanding of the methods, literature, and solutions presented, is truly encouraging.

It is reassuring to know that our emphasis on understanding the semantics of tables and the ontology-driven approach to semantic analysis has been well-received. Your recognition of the potential and reliability of our iterative algorithm, advanced techniques, and the resultant software application confirms the value of our contributions to the field.

Your comment about the paper being suitable for acceptance without any revision is heartening, and we appreciate the trust you have shown in our research.

We would like to take this opportunity to thank you for your time and the professionalism you exhibited during the review process. We understand the effort it takes to thoroughly evaluate a manuscript, and we sincerely appreciate your constructive feedback.

Looking forward to contributing further to the scientific community and remaining open to any additional suggestions or insights in the future.

Warm regards,

Madina Mansurova

Reviewer 5 Report

The context of the research is clearly described and is appropriate to the topic of the research. The foundation of the work is coherently explained in the introduction, however I consider that the number of references used both in the complete work (23 references) and in the introduction and related works section (where only 20 references are used). Included below are some sections that should be improved, justifying what is mentioned in them based on scientific literature:

- About what was mentioned in lines 62-66: “The scientific novelty of this work lies in the creation of a new (automated) method for extracting data from tables in any languages in an arbitrary field of knowledge - subject to the availability of an ontology that describes the data structure. Approaches are also 64 used to classify data more accurately (such as search by cosine distance and a neural network to classify the subject of a table) in comparison with similar algorithms” it is necessary to justify what this statement is based on with some bibliographic reference or work previous.

- In the section on lines 141-144: “A tool called Table2OWL has been developed to transform tables into OWL ontologies. It provides a semi-automatic way to convert tabular data to semantic web formats [13], where manual intervention is required, which can be time consuming, and errors are possible when converting tables in OWL ontologies” it is recommended to explain because it may take more time and what are the possible errors that can happen to tables in OWL ontologies.

- With respect to what was mentioned in the final part of the section in lines 224-226: “Hybrid approaches can be more effective than separate methods because they combine the advantages of each method and achieve more accurate and complete data extraction from tables” it is recommended provide greater depth and justification of why hybrid approaches can be more effective than separate methods, since what is mentioned is quite scarce and poorly founded.

The general research objective is clearly defined and the procedures carried out to achieve it are described. However, there is no mention or reference to the research questions, and their inclusion and link to the purpose of the research is recommended.

Regarding the methodological section, on line 264, mention the table number, since only the title of the table is mentioned. The article presented does not represent an empirical research work, but rather the implementation of an implementation, with each of the materials and procedures to be carried out being adequately and clearly described. The example shown in the results is explanatory and reflects the benefits of the implemented algorithm. However, it is recommended to include a practical example, including the formulation of the evaluation calculation using accuracy, precision, recovery, together with the F1 measurement metric.

In reference to the discussion, given what was mentioned in lines 697-702: “The iterative algorithm developed in this study represents a significant advance in the field of table semantic analysis. Using an ontology-based approach, the algorithm can perform an initial analysis of the table, determining the content type of each cell, and then refine its type based on the ontology. This method ensures accurate data extraction and facilitates the addition of new data to the ontology, improving the analysis of subsequent tables." It is recommended to justify the progress involved in performing an initial analysis of a table, with respect to previous work, determining the type of content of each cell and then refine its type based on the ontology, guaranteeing accurate data extraction, mentioning what difference this process makes with previous works.

On the other hand, it is recommended to give greater depth, both to the limitations, since they are mentioned very briefly, to guarantee the replicability of the work by other researchers, and to the future perspective, mentioning not only the future lines of work. in other contexts, but to justify and discuss the possible lines of research that arise from this work.

Author Response

Dear Reviewer,

We would like to express our sincere gratitude for taking the time to review our manuscript and for your constructive comments and suggestions. Your feedback has been invaluable in improving the quality of our work. 

Our answers are given in the uploaded file, please check the attached file.

Warm regards,

Madina Mansurova

Reviewer 6 Report

This report introduces a promising research endeavor focusing on semantic table analysis using OWL ontology, addressing the increasingly important task of extracting structured information from tables. The work's innovation lies in its adaptability across diverse knowledge domains through ontology-based approaches, which can enhance data classification accuracy. However, while ontologies offer precise semantics, they can be complex to develop and domain-specific. The report's hybrid approach combining rules and ontologies appears well-suited to optimize information extraction. Overall, this research shows great potential for advancing table analysis methods and facilitating data integration with ontologies, though challenges of ontology creation and complexity should be acknowledged.

Author Response

Dear Reviewer,

Thank you for the thoughtful review and constructive feedback on our report. We genuinely appreciate your understanding of the significance and novelty of our work on semantic table analysis utilizing OWL ontology.

Your recognition of our work's innovation, especially its adaptability across a range of knowledge domains and its potential to amplify data classification accuracy, is very encouraging. We completely agree with your observation on the challenges associated with ontology. Indeed, while ontologies provide a structured way to represent knowledge, they do come with complexities and the constraints of being domain-specific. 

We acknowledge the challenges you highlighted, particularly concerning ontology creation and its inherent complexity. 

Once again, thank you for your invaluable feedback. We are open to any further suggestions or insights you might wish to share.

Warm regards,

Madina Mansurova

Round 2

Reviewer 1 Report

The authors addressed the suggestions well. The article can be accepted.

Reviewer 2 Report

The revised paper is suitable for publication.